# Tackling Time-Series Forecasting Generalization via Mitigating Concept Drift

**Zhiyuan Zhao**
Georgia Institute of Technology
leozhao1997@gatech.edu

**Haoxin Liu**
Georgia Institute of Technology
hliu763@gatech.edu

**B. Aditya Prakash**
Georgia Institute of Technology
badityap@cc.gatech.edu

## Abstract

Time-series forecasting finds broad applications in real-world scenarios. Due to the dynamic nature of time series data, it is important for time-series forecasting models to handle potential distribution shifts over time. In this paper, we initially identify two types of distribution shifts in time series: concept drift and temporal shift. We acknowledge that while existing studies primarily focus on addressing temporal shift issues in time series forecasting, designing proper concept drift methods for time series forecasting has received comparatively less attention.

Motivated by the need to address potential concept drift, while conventional concept drift methods via invariant learning face certain challenges in time-series forecasting, we propose a soft attention mechanism that finds invariant patterns from both lookback and horizon time series. Additionally, we emphasize the critical importance of mitigating temporal shifts as a preliminary to addressing concept drift. In this context, we introduce `ShifTS`, a method-agnostic framework designed to tackle temporal shift first and then concept drift within a unified approach. Extensive experiments demonstrate the efficacy of `ShifTS` in consistently enhancing the forecasting accuracy of agnostic models across multiple datasets, and outperforming existing concept drift, temporal shift, and combined baselines.

## 1 Introduction

Time-series forecasting finds applications in various real-world scenarios such as economics, urban computing, and epidemiology (Zhu & Shasha, 2002; Zheng et al., 2014; Deb et al., 2017; Mathis et al., 2024). These applications involve predicting future trends or events based on historical time-series data. For example, economists use forecasts to make financial and marketing plans, while sociologists use them to allocate resources and formulate policies for traffic or disease control.

The recent advent of deep learning has revolutionized time-series forecasting, resulting in a series of advanced forecasting models (Lai et al., 2018; Torres et al., 2021; Salinas et al., 2020; Nie et al., 2023; Zhou et al., 2021). However, despite these successes, time-series forecasting faces certain challenges from distribution shifts due to the dynamic and complex nature of time series data. The distribution shifts in time series can be categorized into two types (Granger, 2003). First, the data distributions of the time series data themselves can change over time, including shifts in mean, variance, and autocorrelation structure, which is referred to as non-stationarity or temporal drift issues in time-series forecasting (Shimodaira, 2000; Du et al., 2021). Second, time-series forecasting is compounded by unforeseen exogenous factors, which shifts the distribution of target time series. These types of phenomena, categorized as concept drift problems in time-series forecasting (Gama et al., 2014; Lu et al., 2018), make it even more challenging.

While prior research has investigated strategies to mitigate temporal shifts (Liu et al., 2022; Kim et al., 2021; Fan et al., 2023), addressing concept drift issues in time-series forecasting has been largely overlooked. Although concept drift is a well-studied problem in general machine learning (Sagawa

et al., 2019; Arjovsky et al., 2019; Ahuja et al., 2021), adapting these solutions to time-series forecasting is challenging. Many of these methods require environment labels, which are typically unavailable in time-series datasets (Liu et al., 2024a). Indeed, the few concept drift approaches developed for time-series data are designed exclusively for online settings (Guo et al., 2021), which requires iterative retraining over time steps and is infeasible when applied to standard time-series forecasting tasks.

Therefore, we aim to close this gap in the literature in this paper, that is, to mitigate concept drift in time-series forecasting for standard time-series forecasting tasks. The contributions of this paper are:

1. **Concept Drift Method:** We introduce soft attention masking (SAM) designed to mitigate concept drift by using the invariant patterns in exogenous features. The soft attention allows the time-series forecasting models to weigh and ensemble of invariant patterns at multiple horizon time steps to enhance the generalization ability.

2. **Distribution Shift Generalized Framework:** We show the necessity of addressing temporal shift as a preliminary when addressing concept drift. We therefore propose ShifTS, a practical, distribution shift generalized, model-agnostic framework that tackles temporal shift and concept drift within a unified approach.

3. **Comprehensive Evaluations:** We conduct extensive experiments on various time series datasets with multiple advanced time-series forecasting models. The proposed ShifTS demonstrates effectiveness by consistent performance improvements to agnostic forecasting models, as well as outperforming distribution shift baselines in better forecasting accuracy.

We provide related works on time-series analysis and distribution shift generalization in Appendix A.

## 2 PROBLEM FORMULATION

### 2.1 TIME-SERIES FORECASTING

Time-series forecasting involves predicting future values of one or more dependent time series based on historical data, augmented with exogenous covariate features. Let denote the target time series as $\mathbf{Y}$ and its associated exogenous covariate features as $\mathbf{X}$. At any time step $t$, time-series forecasting aims to predict $\mathbf{Y}_t^H = [yt + 1, y_{t+2}, \ldots, y_{t+H}] \in \mathbf{Y}$ using historical data $(\mathbf{X}_t^L, \mathbf{Y}_t^L)$, where $L$ represents the length of the historical data window, known as the *lookback window*, and $H$ denotes the forecasting time steps, known as the *horizon window*. Here, $\mathbf{X}_t^L = [x_{t-L+1}, x_{t-L+2}, \ldots, x_t] \in \mathbf{X}$ and $\mathbf{Y}_t^L = [y_{t-L+1}, y_{t-L+2}, \ldots, y_t] \in \mathbf{Y}$. For simplicity, we denote $\mathbf{Y}^H = \{\mathbf{Y}_t^H\}$ for $\forall t$ as the collection of horizon time-series of all time steps, and similar for $\mathbf{Y}^L$ and $\mathbf{X}^L$. Conventional time-series forecasting involves learning a model parameterized by $\theta$ through empirical risk minimization (ERM) to obtain $f_\theta : (\mathbf{X}^L, \mathbf{Y}^L) \to \mathbf{Y}^H$ for all time steps $t$. In this study, we focus on univariate time-series forecasting with exogenous features, where $d_{\mathbf{Y}} = 1$ and $d_{\mathbf{X}} \geq 1$.

### 2.2 DISTRIBUTION SHIFT IN TIME SERIES

Given the time-series forecasting setups, a time-series forecasting model aims to predict the target distribution $\mathrm{P}(\mathbf{Y}^H) = \mathrm{P}(\mathbf{Y}^H|\mathbf{Y}^L)\mathrm{P}(\mathbf{Y}^L) + \mathrm{P}(\mathbf{Y}^H|\mathbf{X}^L)\mathrm{P}(\mathbf{X}^L)$, which should be generalizable for both training and testing time steps. However, due to the dynamic nature of time-series data, forecasting faces challenges from distribution shifts, categorized into two types: temporal shift and concept drift. These two types of distribution shifts are defined as follows:

**Definition 2.1 (Temporal Shift (Shimodaira, 2000; Du et al., 2021))** *Temporal shift (also known as virtual shift (Tsymbal, 2004)) refers to phenomenon that the marginal probability distributions change over time, while the conditional distributions are the same.*

**Definition 2.2 (Concept Drift (Lu et al., 2018))** *Concept drift (also known as real concept drift (Gama et al., 2014)[1]) is the phenomenon where the conditional distributions change over time, while the marginal probability distributions are the same.*

---

[1](Gama et al., 2014) defines concept drift as both virtual shift and real concept drift. Our concept drift definition is consistent with the definition of real concept drift in (Gama et al., 2014).

Intuitively, a temporal shift indicates unstable marginal distributions (e.g. $P(\mathbf{Y}^H) \neq P(\mathbf{Y}^L)$), while a concept drift indicates unstable conditional distributions ($P(\mathbf{Y}_i^H|\mathbf{X}_i^L) \neq P(\mathbf{Y}_j^H|\mathbf{X}_j^L)$ for some $i, j \in t$). Existing methods for distribution shifts in time-series forecasting typically focus on mitigating temporal shifts through normalization, ensuring $P(\mathbf{Y}^H) = P(\mathbf{Y}^L)$ by both normalizing to standard 0-1 distributions (Kim et al., 2021; Liu et al., 2022; Fan et al., 2023). In contrast, concept drift remains relatively underexplored in time-series forecasting.

Nevertheless, time-series forecasting does face challenges from concept drift: The correlations between $\mathbf{X}$ and $\mathbf{Y}$ can change over time, making the conditional distributions $P(\mathbf{Y}^H|\mathbf{X}^L)$ unstable and less predictable. A demonstration visualizing the differences and relationships between temporal shift and concept drift is provided in Appendix B.

While the concept drift issue has received considerable attention in existing studies on general machine learning, applying them, mostly invariant learning approaches, to time-series forecasting tasks presents certain challenges. Firstly, conventional approaches to mitigate concept drift are through invariant learning. However, these invariant learning methods typically rely on explicit environment labels as input (e.g., labeled rotation or noisy images in image classification), which are not readily available in time series datasets. Second, these invariant learning methods assume that all correlated exogenous features necessary to fully determine the target variable are accessible (Liu et al., 2024a), which are often not applied to time series datasets (e.g., lookback window information is not sufficiently determining the horizon target). Indeed, a few concept drift methods not based on invariant learning have been proposed for time-series forecasting (Guo et al., 2021). However, these methods are designed for the online setting which does not fit standard time-series forecasting, and are only validated on limited synthetic datasets rather than complicated real-world ones.

## 3 METHODOLOGY

The main idea of our methodology is to address concept drift through SAM by modeling stable conditional distributions on surrogate exogenous features with invariant patterns, rather than the sole lookback window. Furthermore, we recognize that effectively mitigating temporal shifts is preliminary for addressing concept drift. To this end, we propose ShifTS that effectively handles concept drift by first resolving temporal shifts as a preliminary step within a unified framework.

### 3.1 MITIGATING CONCEPT DRIFT

**Methodology Intuition.** As defined in Definition 2.2, concept drift in time-series refers to the changing correlations between $\mathbf{X}$ and $\mathbf{Y}$ over time ($P(\mathbf{Y}_i^H|\mathbf{X}_i^L) \neq P(\mathbf{Y}_j^H|\mathbf{X}_j^L)$ for $i, j \in t$), which introduces instability when modeling conditional distribution $P(\mathbf{Y}^H|\mathbf{X}^L)$.

This instability arises because, for a given exogenous feature $\mathbf{X}$, its lookback window $\mathbf{X}^L$ alone may lack sufficient information to predict $\mathbf{Y}^H$, while learning a stable conditional distribution requires that the inputs provide sufficient information to predict the output (Sagawa et al., 2019; Arjovsky et al., 2019). There are possible patterns in the horizon window $\mathbf{X}^H$, joint with $\mathbf{X}^L$, that influence the target. Thus, modeling $P(\mathbf{Y}^H|\mathbf{X}^L, \mathbf{X}^H)$ leads to a more stable conditional distribution compared to $P(\mathbf{Y}^H|\mathbf{X}^L)$, as $[\mathbf{X}^L, \mathbf{X}^H]$ captures additional causal relationships across future time steps. We assume that incorporating causal relationships from the horizon window enables more complete causality modeling between that exogenous feature and target, given that the future cannot influence the past (e.g., $\mathbf{X}_{t+1}^H \nrightarrow \mathbf{Y}_t^H$). However, these causal effects from the horizon window, while important for learning stable conditional distributions, are often overlooked by conventional time-series forecasting methods, as illustrated in Figure 1(a).

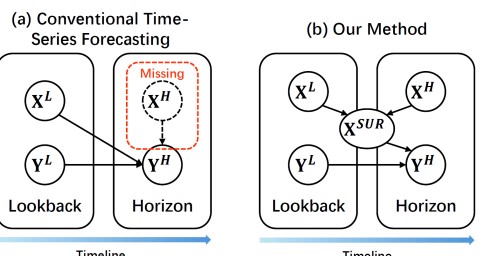

Figure 1: Comparison between conventional time-series forecasting and our approach. Our approach identifies invariant patterns in lookback and horizon window as $\mathbf{X}^{SUR}$ and then models a stable conditional distribution accordingly to mitigate concept drift.

Therefore, we propose leveraging both lookback and horizon information from exogenous features (i.e., $[\mathbf{X}^L, \mathbf{X}^H]$) to predict the target, enabling a more stable conditional distribution. However, directly modeling $P(\mathbf{Y}^H|\mathbf{X}^L, \mathbf{X}^H)$ in practice presents two challenges. First, $\mathbf{X}^H$ typically represents unknown future values during testing. To model $P(\mathbf{Y}^H|\mathbf{X}^L, \mathbf{X}^H)$, it may require to first predict $\mathbf{X}^H$ by modeling $P(\mathbf{X}^H|\mathbf{X}^L)$, which can be as challenging as predicting $\mathbf{Y}^H$ directly. Second, not every pattern in $\mathbf{X}^H$ at every time step holds a causal relationship with the target. Modeling all patterns from $\mathbf{X}^L$ and $\mathbf{X}^H$ may introduce noisy causal relationships (as invariant learning methods aim to mitigate) and reduce the stability of conditional distributions.

To address the above challenges, instead of directly modeling $P(\mathbf{Y}^H|\mathbf{X}^L, \mathbf{X}^H)$, we propose a two-step approach: first, identifying patterns in $[\mathbf{X}^L, \mathbf{X}^H]$ that lead to stable conditional distributions (namely invariant patterns), and then modeling these conditional distributions accordingly. To determine stability, a natural intuition is to assess whether a pattern's correlation with the target remains consistent across all time steps. For instance, if a subsequence of $[\mathbf{X}^L, \mathbf{X}^H]$ consistently exhibits stable correlations with the target over all or most time steps (e.g., an increase of the subsequence always results in an increase of the target), then its conditional distribution should be explicitly modeled due to the stability. Conversely, if a subsequence demonstrates correlations with the target only sporadically or locally, these correlations are likely spurious, which are unstable conditional distributions to other time steps. We leverage this intuition to identify all invariant patterns and aggregate them into a surrogate feature $\mathbf{X}^{\text{SUR}}$, accounting for the fact that the target can be determined by multiple patterns. For instance, an influenza-like illness (ILI) outbreak in winter can be triggered by either extreme cold weather in winter or extreme heat waves in summer (Nielsen et al., 2011; Jaakkola et al., 2014). By incorporating this information, we model the corresponding conditional distribution $P(\mathbf{Y}^H|\mathbf{X}^{\text{SUR}})$, as illustrated in Figure 1(b).

The effectiveness of $\mathbf{X}^{\text{SUR}}$ in predicting $\mathbf{Y}^H$ stems from two key insights. First, $P(\mathbf{Y}^H|\mathbf{X}^{\text{SUR}})$ is a stable conditional distribution to model, as it captures invariant patterns across both the lookback and horizon windows. Second, while there is a trade-off—$P(\mathbf{Y}^H|\mathbf{X}^{\text{SUR}})$ provides stability, but estimating $\mathbf{X}^{\text{SUR}}$ may introduce additional errors—practical evaluations demonstrate that the benefits of constructing stable conditional distributions outweigh the potential estimation errors of $\mathbf{X}^{\text{SUR}}$. This is because $\mathbf{X}^{\text{SUR}}$ contains only partial information, which is easier to predict than the entire $\mathbf{X}^H$.

**Methodology Implementation.** Recognizing that $P(\mathbf{Y}^H|\mathbf{X}^{SUR})$ is the desirable conditional distribution to learn, the remaining challenge is to identify $\mathbf{X}^{SUR}$ in practice. To achieve this, we propose a soft attention masking mechanism (SAM), that operates as follows: First, we concatenate $[\mathbf{X}^L, \mathbf{X}^H]$ to form an entire time series of length $L + H$. The entire series is then sliced using a sliding window of size $H$, resulting in $L + 1$ slices. This process extracts local patterns ($[\mathbf{X}_{t-L}^H, \ldots, \mathbf{X}_t^H]$ at each time step $t$), which are subsequently used to identify invariant patterns.

Second, we model the conditional distributions for all local patterns $[P(\mathbf{Y}_t^H|\mathbf{X}_{t-L}^H), \ldots, P(\mathbf{Y}_t^H|\mathbf{X}_t^H)]$ at each time step $t$, with applying a learnable soft attention matrix $\mathcal{M}$ to weigh each local pattern. This matrix incorporates softmax, sparsity, and normalization operations, which can be mathematically described as:

$$
\begin{aligned}
\text{Softmax:} \quad & \mathcal{M}_j = \text{Softmax}(\mathcal{M}_j) \\
\text{Sparsity:} \quad & \mathcal{M}_{ij} = \mathcal{M}_{ij} \cdot \mathbb{1}_{(\mathcal{M}_{ij} - \mu(\mathcal{M}_j)) \geq 0} \\
\text{Normalize:} \quad & \mathcal{M}_j = \frac{\mathcal{M}_j}{|\mathcal{M}_j|}
\end{aligned}
\tag{1}
$$

where $i, j$ are the first and second dimensions of $\mathcal{M}$. These operations are essential for SAM identifying invariant patterns. The intuition is that we consider sliced windows from the lookback and horizon over time steps as candidates of invariant patterns. We use the softmax operation to compute and update the weights of each pattern contributing to the target $\mathbf{Y}^H$. We then apply a sparsity operation to filter out patterns with low weights, leaving only the patterns with high weights. These high-weight patterns, which consistently contribute to the target across all instances at all time steps, are regarded as invariant patterns over time. These patterns intuitively are invariant patterns as $P(\mathbf{Y}_i^H|\mathbf{X}_{i-k}^H) \approx P(\mathbf{Y}_j^H|\mathbf{X}_{j-k}^H)$ for some $k \in [0, L]$ and $i, j \in t$. While multiple invariant patterns may be identified, we compute a weighted sum of these patterns, proportional to their contributions

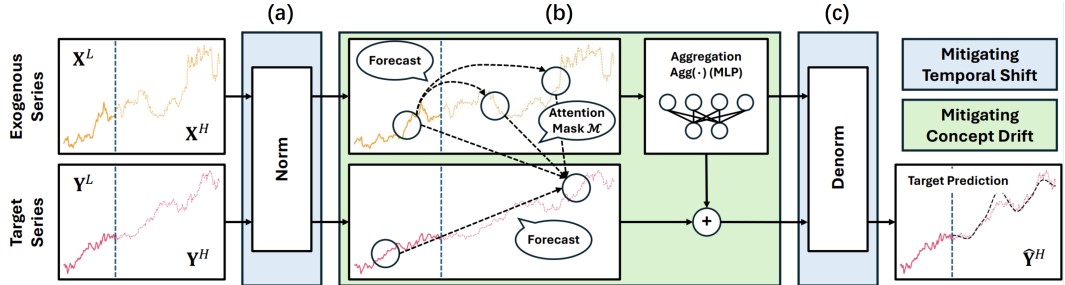

Figure 2: Diagram of ShifTS, consisting of three components: (a) normalization at the start (c) denormalization at the end to address temporal shifts, and (b) a two-stage forecasting process-The first stage predicts surrogate exogenous features, $\hat{\mathbf{X}}^{\text{SUR}}$, identified by the SAM, which capture invariant patterns essential for forecasting the target; The second stage uses both the predicted surrogate exogenous features and the original $\mathbf{Y}^L$ to predict $\mathbf{Y}^H$.

in predicting the target. The weighted-sum patterns formulate the surrogate feature $\mathbf{X}^{SUR}$. For simplicity, we denote this process as:

$$\mathbf{X}^{\text{SUR}} = \text{SAM}([\mathbf{X}^L, \mathbf{X}^H]) = \sum_{L+1} \mathcal{M}(\text{Slice}([\mathbf{X}^L, \mathbf{X}^H])) \tag{2}$$

where $\text{Slice}(\cdot)$ represents slicing the time series $[L + H, d_{\mathbf{X}}] \to [H, L + 1, d_{\mathbf{X}}]$, and $\mathcal{M} \in \mathbb{R}^{L+1 \times d_{\mathbf{X}}}$ is the learnable soft attention as in Equation 1.

In practice, $\mathbf{X}^{\text{SUR}}$ may include horizon information unavailable during testing. To address this, SAM estimates the surrogate features $\hat{\mathbf{X}}^{\text{SUR}}$ using agnostic forecasting models. The surrogate loss that aims to estimate $\hat{\mathbf{X}}^{\text{SUR}}$ is defined as:

$$\mathcal{L}_{\text{SUR}} = \text{MSE}(\mathbf{X}^{\text{SUR}}, \hat{\mathbf{X}}^{\text{SUR}}) \tag{3}$$

## 3.2 MITIGATING TEMPORAL SHIFT

While the primary contribution of this work is to mitigate concept drift in time-series forecasting, addressing temporal shifts is equally critical and serves as a prerequisite for effectively managing concept drift. The key intuition is that SAM seeks to learn invariant patterns that result in a stable conditional distribution, $P(\mathbf{Y}^H|\mathbf{X}^{\text{SUR}})$. However, achieving this stability becomes challenging if the marginal distributions (e.g., $P(\mathbf{Y}^H)$ or $P(\mathbf{X}^{\text{SUR}})$) are not fixed, as these distributions may change over time because of the temporal shift issues.

To address this issue, a natural solution is to learn the conditional distribution under standardized marginal distributions. This can be achieved using temporal shift methods, which employ instance normalization techniques to stabilize the marginals. The core intuition behind popular temporal shift methods is to normalize data distributions before the model processes them and to denormalize the outputs afterward. This approach ensures that the normalized sequences maintain consistent mean and variance between the inputs and outputs of the forecasting model. Specifically, $P(\mathbf{X}^L_{\text{Norm}}) \approx P(\mathbf{X}^H_{\text{Norm}}) \sim \text{Dist}(0,1)$ and $P(\mathbf{Y}^L_{\text{Norm}}) \approx P(\mathbf{Y}^H_{\text{Norm}}) \sim \text{Dist}(0,1)$, thereby mitigating temporal shifts (i.e., shifts in marginal distributions over time).

Among the existing methods, Reversible Instance Normalization (RevIN) (Kim et al., 2021) stands out for its simplicity and effectiveness, making it the method of choice in this work. Advanced techniques, such as SAN (Liu et al., 2023) and N-S Transformer (Liu et al., 2022), have also demonstrated promise in addressing temporal shifts. However, these methods often require modifications to forecasting models or additional pre-training strategies. While exploring these advanced temporal shift approaches remains a promising avenue for further performance improvements, it is beyond the scope of this study and not the primary focus of this work.

### 3.3 SHIFTS: THE INTEGRATED FRAMEWORK

To address concept drift in time-series forecasting, while acknowledging that mitigating temporal shifts is a prerequisite for resolving concept drift, we propose ShifTS —a comprehensive framework designed to tackle both challenges in time-series forecasting. ShifTS is model-agnostic, as the stable conditional distributions distinguished by SAM can be learned by any time-series forecasting model. The workflow of ShifTS is illustrated in Figure 2 and consists of the following steps: (1) Normalize the input time series; (2) Forecast surrogate exogenous features $\hat{\mathbf{X}}^{\mathrm{SUR}}$ that invariantly support the target series, as determined by SAM; (3) An aggregation MLP that uses $\hat{\mathbf{X}}^{\mathrm{SUR}}$ to forecast the target, denoted as $\mathrm{Agg}(\cdot)$ in Figure 2 and Algorithm 1; (4) Denormalize the output time series. Conceptually, steps 1 and 4 mitigate the temporal shift, step 2 addresses concept drift, and step 3 performs weighted aggregation of exogenous features to support the target series. The optimization objective of ShifTS is as follows:

$$\mathcal{L} = \mathcal{L}_{\mathrm{SUR}}(\mathbf{X}^{\mathrm{SUR}}, \hat{\mathbf{X}}^{\mathrm{SUR}}) + \mathcal{L}_{\mathrm{TS}}(\mathbf{Y}^H, \hat{\mathbf{Y}}^H) \tag{4}$$

Here, $\mathcal{L}_{\mathrm{SUR}}$ is the surrogate loss that encourages learning to forecast exogenous features, and $\mathcal{L}_{\mathrm{TS}}$ is the MSE loss used in conventional time-series forecasting. The pseudo-code for training and testing ShifTS is provided in Algorithm 1.

---

**Algorithm 1** ShifTS

---

1: **Training: Require:** Training data $\mathbf{X}^L$, $\mathbf{X}^H$, $\mathbf{Y}^L$, $\mathbf{Y}^H$; Initial parameters $f_0$, $\mathcal{M}_0$, $\mathrm{Agg}_0$; **Output:** Model parameter $f$, $\mathcal{M}$, $\mathrm{Agg}$

2: For $i$ in range ($E$):
3:     Normalization:      $[\mathbf{X}^L_{\mathrm{Norm}}, \mathbf{Y}^L_{\mathrm{Norm}}] = \mathrm{Norm}([\mathbf{X}^L, \mathbf{Y}^L])$
4:     Time-series forecasting:      $[\hat{\mathbf{X}}^{\mathrm{SUR}}_{\mathrm{Norm}}, \hat{\mathbf{Y}}^H_{\mathrm{Norm}}] = f_i([\mathbf{X}^L_{\mathrm{Norm}}, \mathbf{Y}^L_{\mathrm{Norm}}])$
5:     Exogenous feature aggregation:      $\hat{\mathbf{Y}}^H_{\mathrm{Norm}} = \hat{\mathbf{Y}}^H_{\mathrm{Norm}} + \mathrm{Agg}_i(\hat{\mathbf{X}}^{\mathrm{SUR}}_{\mathrm{Norm}})$
6:     Denormalization:      $[\hat{\mathbf{X}}^{\mathrm{SUR}}, \hat{\mathbf{Y}}^H] = \mathrm{Denorm}([\hat{\mathbf{X}}^{\mathrm{SUR}}_{\mathrm{Norm}}, \hat{\mathbf{Y}}^H_{\mathrm{Norm}}])$
7:     Obtain sufficient ex-features:      $\mathbf{X}^{\mathrm{SUR}} = \mathrm{SAM}([\mathbf{X}^L, \mathbf{X}^H])$
8:     Compute loss:      $\mathcal{L} = \mathcal{L}_{\mathrm{SUR}}(\mathbf{X}^{\mathrm{SUR}}, \hat{\mathbf{X}}^{\mathrm{SUR}}) + \mathcal{L}_{\mathrm{TS}}(\mathbf{Y}^H, \hat{\mathbf{Y}}^H)$
9:     Update model parameter:      $f_{i+1} \leftarrow f_i, \mathcal{M}_{i+1} \leftarrow \mathcal{M}_i, \mathrm{Agg}_{i+1} \leftarrow \mathrm{Agg}_i$
10: Final model parameters: $f \leftarrow f_E, \mathcal{M} \leftarrow \mathcal{M}_E, \mathrm{Agg} \leftarrow \mathrm{Agg}_E$

11: **Testing: Require:** Test data $\mathbf{X}^L$, $\mathbf{Y}^L$, **Output:** Forecast target $\hat{\mathbf{Y}}^H$

12:     Normalization:      $[\mathbf{X}^L_{\mathrm{Norm}}, \mathbf{Y}^L_{\mathrm{Norm}}] = \mathrm{Norm}([\mathbf{X}^L, \mathbf{Y}^L])$
13:     Time-series forecasting:      $[\hat{\mathbf{X}}^{\mathrm{SUR}}_{\mathrm{Norm}}, \hat{\mathbf{Y}}^H_{\mathrm{Norm}}] = f([\mathbf{X}^L_{\mathrm{Norm}}, \mathbf{Y}^L_{\mathrm{Norm}}])$
14:     Exogenous feature aggregation:      $\hat{\mathbf{Y}}^H_{\mathrm{Norm}} = \hat{\mathbf{Y}}^H_{\mathrm{Norm}} + \mathrm{Agg}(\hat{\mathbf{X}}^{\mathrm{SUR}}_{\mathrm{Norm}})$
15:     Denormalization:      $[\hat{\mathbf{X}}^{\mathrm{SUR}}, \hat{\mathbf{Y}}^H] = \mathrm{Denorm}([\hat{\mathbf{X}}^{\mathrm{SUR}}_{\mathrm{Norm}}, \hat{\mathbf{Y}}^H_{\mathrm{Norm}}])$

---

## 4 EXPERIMENTS

### 4.1 SETUP

**Datasets.** We conduct experiments using six time-series datasets as leveraged in (Liu et al., 2024a): The daily reported currency exchange rates (**Exchange**) (Lai et al., 2018); The weekly reported influenza-like illness patients (**ILI**) (Kamarthi et al., 2021); Two-hourly/minutely reported electricity transformer temperature (**ETTh1/ETTh2** and **ETTm1/ETTm2**, respectively) (Zhou et al., 2021). We follow the established experimental setups and target variable selections in previous works(Wu et al., 2021; 2022; Nie et al., 2023; Liu et al., 2024d). Datasets such as Traffic (PeMS) (Zhao et al., 2017) and Weather (Wu et al., 2021) are excluded from our evaluations, as their time series exhibit near-stationary behavior, with only moderate distribution shift issues. Further details on the dataset differences are discussed in Appendix C.1.

**Baselines.** We include two types of baselines for comprehensive evaluation on ShifTS:

**Forecasting Model Baselines**: ShifTS is model-agnostic, we include six time-series forecasting models (referred to as 'Model' in Table 1 and 4), including: **Informer** (Zhou et al., 2021),

Table 1: Performance comparison on forecasting errors without (ERM) and with `ShifTS`. Employing `ShifTS` shows consistent performance gains agnostic to forecasting models. The top-performing method is in bold. 'IMP.' denotes the average improvements over all horizons of `ShifTS` vs ERM.

| Model | | Crossformer (ICLR'23) | | | | PatchTST (ICLR'23) | | | | iTransformer (ICLR'24) | | | |
|---|---|---|---|---|---|---|---|---|---|---|---|---|---|
| Method | | ERM | | ShifTS | | ERM | | ShifTS | | ERM | | ShifTS | |
| Dataset | | MSE | MAE | MSE | MAE | MSE | MAE | MSE | MAE | MSE | MAE | MSE | MAE |
| ILI | 24 | 3.409 | 1.604 | **0.674** | **0.590** | 0.772 | 0.634 | **0.656** | **0.618** | 0.824 | 0.653 | **0.799** | **0.642** |
| | 36 | 4.001 | 1.772 | **0.687** | **0.617** | 0.763 | 0.649 | **0.694** | **0.602** | 0.917 | 0.738 | **0.690** | **0.640** |
| | 48 | 3.720 | 1.724 | **0.652** | **0.611** | 0.753 | 0.692 | **0.654** | **0.630** | 0.772 | 0.699 | **0.680** | **0.665** |
| | 60 | 3.689 | 1.715 | **0.658** | **0.633** | 0.761 | 0.724 | **0.680** | **0.656** | 0.729 | 0.710 | **0.672** | **0.667** |
| | IMP. | | | 81.9% | 64.0% | | | 12.0% | 7.1% | | | 13.8% | 6.5% |
| Exchange | 96 | 0.338 | 0.475 | **0.102** | **0.237** | 0.130 | 0.265 | **0.102** | **0.236** | 0.135 | 0.272 | **0.115** | **0.255** |
| | 192 | 0.566 | 0.622 | **0.203** | **0.338** | 0.247 | 0.394 | **0.194** | **0.332** | 0.250 | 0.376 | **0.209** | **0.343** |
| | 336 | 1.078 | 0.867 | **0.407** | **0.484** | 0.522 | 0.557 | **0.388** | **0.477** | 0.450 | 0.503 | **0.426** | **0.495** |
| | 720 | 1.292 | 0.963 | **1.165** | **0.813** | 1.171 | 0.824 | **0.995** | **0.747** | 1.501 | 0.941 | **1.138** | **0.827** |
| | IMP. | | | 53.5% | 38.9% | | | 20.9% | 12.6% | | | 15.2% | 6.9% |
| ETTh1 | 96 | 0.145 | 0.312 | **0.055** | **0.180** | 0.064 | 0.193 | **0.056** | **0.181** | 0.061 | 0.190 | **0.056** | **0.181** |
| | 192 | 0.240 | 0.420 | **0.072** | **0.206** | 0.085 | 0.222 | **0.073** | **0.209** | 0.076 | 0.219 | **0.072** | **0.205** |
| | 336 | 0.240 | 0.424 | **0.084** | **0.228** | 0.096 | 0.244 | **0.089** | **0.235** | 0.086 | 0.227 | **0.083** | **0.225** |
| | 720 | 0.391 | 0.553 | **0.095** | **0.244** | 0.128 | 0.282 | **0.097** | **0.245** | 0.085 | 0.232 | **0.082** | **0.230** |
| | IMP. | | | 68.2% | 48.8% | | | 14.5% | 7.2% | | | 5.1% | 3.3% |
| ETTh2 | 96 | 0.255 | 0.408 | **0.137** | **0.286** | 0.154 | 0.309 | **0.139** | **0.287** | 0.141 | 0.292 | **0.137** | **0.288** |
| | 192 | 1.257 | 1.034 | **0.182** | **0.338** | 0.204 | 0.374 | **0.191** | **0.345** | 0.194 | 0.347 | **0.184** | **0.339** |
| | 336 | 0.783 | 0.771 | **0.234** | **0.388** | 0.252 | 0.406 | **0.222** | **0.381** | 0.229 | 0.383 | **0.225** | **0.381** |
| | 720 | 1.455 | 1.100 | **0.234** | **0.389** | 0.259 | 0.411 | **0.236** | **0.390** | 0.266 | 0.413 | **0.235** | **0.390** |
| | IMP. | | | 71.4% | 52.9% | | | 9.2% | 6.5% | | | 5.4% | 2.5% |
| ETTm1 | 96 | 0.050 | 0.174 | **0.028** | **0.126** | 0.031 | 0.135 | **0.029** | **0.128** | **0.030** | **0.131** | **0.030** | **0.131** |
| | 192 | 0.271 | 0.454 | **0.043** | **0.158** | 0.048 | 0.166 | **0.044** | **0.161** | 0.049 | 0.171 | **0.046** | **0.165** |
| | 336 | 0.731 | 0.805 | **0.057** | **0.184** | **0.058** | 0.190 | **0.058** | **0.186** | 0.066 | 0.199 | **0.059** | **0.188** |
| | 720 | 0.829 | 0.849 | **0.083** | **0.219** | 0.083 | 0.223 | **0.080** | **0.219** | 0.082 | 0.219 | **0.079** | **0.217** |
| | IMP. | | | 77.3% | 61.0% | | | 4.6% | 3.0% | | | 5.1% | 2.5% |
| ETTm2 | 96 | 0.153 | 0.315 | **0.069** | **0.190** | 0.078 | 0.206 | **0.067** | **0.188** | **0.073** | **0.200** | **0.073** | **0.195** |
| | 192 | 0.408 | 0.526 | **0.105** | **0.242** | 0.113 | 0.246 | **0.101** | **0.237** | 0.119 | 0.251 | **0.108** | **0.248** |
| | 336 | 0.428 | 0.504 | **0.146** | **0.289** | 0.176 | 0.320 | **0.134** | **0.278** | 0.157 | 0.302 | **0.144** | **0.291** |
| | 720 | 1.965 | 1.205 | **0.191** | **0.342** | 0.220 | 0.368 | **0.185** | **0.334** | 0.196 | 0.347 | **0.193** | **0.344** |
| | IMP. | | | 71.3% | 52.0% | | | 15.9% | 8.6% | | | 4.8% | 2.1% |

**Pyraformer** (Liu et al., 2021), **Crossformer** (Zhang & Yan, 2022), **PatchTST** (Nie et al., 2023), **TimeMixer** (Wang et al., 2024) and **iTransformer** (Liu et al., 2024d), which of the last two are the state-of-the-art (SOTA) forecasting model. These models are used to demonstrate that `ShifTS` consistently enhances forecasting accuracy across various models, including SOTA.

**Distribution Shift Baselines**: We compare `ShifTS` with various distribution shift methods (referred to as 'Method' in Table 2): (1) Three non-stationary methods for addressing temporal distribution shifts in time-series forecasting **N-S Trans.** (Liu et al., 2022), **RevIN** (Kim et al., 2021), and **SAN** (Liu et al., 2023). We omit **Dish-TS** (Fan et al., 2023) and **SIN** (Han et al., 2024) from the main text due to their instability on univariate targets. (2) Four concept drift methods, including **GroupDRO** (Sagawa et al., 2019), **IRM** (Arjovsky et al., 2019), **VREx** (Krueger et al., 2021), and **EIIL** (Creager et al., 2021), which are primarily designed for general applications. (3) Three combined methods for both temporal distribution shifts and concept drift: **IRM+RevIN**, **EIIL+RevIN**, and SOTA time-series distribution shift method **FOIL** (Liu et al., 2024a). These comparisons aim to highlight the advantages of `ShifTS` in distribution shift generalization over existing distribution shift approaches.

**Evaluation.** We measure the forecasting errors using mean squared error (**MSE**) and mean absolute error (**MAE**). The formula of the metrics are: $\text{MSE} = \frac{1}{n}\sum_{i=1}^{n}(\boldsymbol{y}-\hat{\boldsymbol{y}})^2$ and $\text{MAE} = \frac{1}{n}\sum_{i=1}^{n}|\boldsymbol{y}-\hat{\boldsymbol{y}}|$.

**Reproducibility.** All models are trained on NVIDIA Tesla V100 32GB GPUs. All training data and code are available at: `https://github.com/AdityaLab/ShifTS`. More experiment details are presented in Appendix C.2.

## 4.2 PERFORMANCE IMPROVEMENT ACROSS BASE FORECASTING MODELS

To evaluate the effectiveness of `ShifTS` in reducing forecasting errors, we conduct experiments comparing performance with and without `ShifTS` across popular time-series datasets and four

Table 2: Averaged performance comparison between `ShifTS` and distribution shift baselines with Crossformer. `ShifTS` achieves the best and second-best performance in 6 and 2 out of 8 evaluations. The best results are highlighted in bold and the second-best results are underlined.

| Dataset | | ILI | | Exchange | | ETTh1 | | ETTh2 | |
|---|---|---|---|---|---|---|---|---|---|
| Method | | MSE | MAE | MSE | MAE | MSE | MAE | MSE | MAE |
| Base | ERM | 3.705 | 1.704 | 0.819 | 0.732 | 0.254 | 0.427 | 0.937 | 0.828 |
| Concept Drift Method | GroupDRO | 2.285 | 1.287 | 0.821 | 0.751 | 0.278 | 0.453 | 1.150 | 0.936 |
| | IRM | 2.248 | 1.237 | 0.846 | 0.754 | 0.201 | 0.367 | 0.878 | 0.792 |
| | VREx | 2.285 | 1.286 | 0.821 | 0.742 | 0.314 | 0.486 | 1.142 | 0.938 |
| | EIIL | 2.036 | 1.159 | 0.822 | 0.749 | 0.212 | 0.433 | 1.122 | 0.930 |
| Temporal Shift Method | RevIN | 0.815 | 0.708 | 0.475 | 0.476 | 0.085 | 0.224 | 0.205 | 0.358 |
| | N-S Trans. | 0.781 | 0.688 | 0.484 | 0.481 | 0.086 | 0.226 | 0.203 | 0.355 |
| | SAN | 0.757 | 0.715 | **0.415** | **0.453** | 0.088 | 0.225 | 0.199 | 0.348 |
| Combined Method | IRM+RevIN | 0.809 | 0.711 | 0.481 | 0.476 | 0.089 | 0.231 | 0.202 | 0.362 |
| | EIIL+RevIN | 0.799 | 0.706 | 0.483 | 0.485 | 0.085 | 0.225 | 0.218 | 0.380 |
| | FOIL | 0.735 | 0.651 | 0.497 | 0.481 | 0.081 | 0.219 | 0.206 | 0.357 |
| | **ShifTS (Ours)** | **0.668** | **0.613** | 0.470 | 0.468 | **0.076** | **0.214** | **0.194** | **0.348** |

different forecasting horizons. These experiments utilize five transformer-based models and one MLP-based model. Evaluation results for Crossformer, PatchTST, and iTransformer are presented in Table 1, while additional results for older models, including Informer, Pyraformer, and TimeMixer, are provided in Table 4 in Appendix D.1.

The experimental results consistently demonstrate the effectiveness of `ShifTS` in improving forecasting performance across agnostic forecasting models. Notably, `ShifTS` achieves reductions in forecasting errors of up to 15% when integrated with advanced models like iTransformer. Furthermore, `ShifTS` shows even greater relative effectiveness when applied to older or less advanced forecasting models, such as Informer and Crossformer.

In addition to the observed performance improvements, our results reveal two further insights:

**The effectiveness of `ShifTS` relies on the insights provided by the horizon data.** The performance improvements exhibit variations across different datasets. For instance, the application of `ShifTS` on ILI and Exchange datasets yields greater performance improvements compared to ETT datasets overall. To interpret the phenomenon and determine the conditions under which `ShifTS` could be most effective in practical scenarios, we quantify the mutual information $I(\mathbf{X}^H; \mathbf{Y}^H)$ shared between $\mathbf{X}^H$ and $\mathbf{Y}^H$ (detailed setup provided in Appendix C.2). We plot the relationship between $I(\mathbf{X}^H; \mathbf{Y}^H)$ and performance gains in Figure 3(a). The scatter plot illustrates a positive linear correlation between $I(\mathbf{X}^H; \mathbf{Y}^H)$ and performance gains, supported by a p-value $p = 0.012 \leq 0.05$. This observation suggests that the greater the amount of useful information from exogenous features within the horizon window, the more substantial the performance gains achieved by `ShifTS`. This insight aligns with the design of `ShifTS`, as higher mutual information indicates clearer correlations and causal relationships between the target $\mathbf{Y}^H$ and exogenous features in the horizon window—relationships often overlooked by conventional time-series models. Stronger correlations imply a greater extent of misrepresented dependencies in ERM, leading to more significant improvements with `ShifTS`.

**The extent of quantitative performance gains achieved by `ShifTS` depends on the underlying forecasting model.** Notably, the extent of performance enhancements achieved by `ShifTS` varies across different forecasting models. For example, the performance gains on the simpler Informer model by `ShifTS` is more significant than the SOTA iTransformer model. Importantly, we emphasize two key observations: Firstly, even when applied to the iTransformer model, `ShifTS` demonstrates a notable performance boost of approximately 15% on both ILI and Exchange datasets, consistent with the aforehead intuition. Secondly, integrating `ShifTS` into forecasting processes should, at the very least, maintain or improve the performance of standalone forecasting models, as evidenced by consistent performance enhancements observed across all datasets with iTransformer model.

## 4.3 COMPARISON WITH DISTRIBUTION SHIFT METHODS

To illustrate the advantages of `ShifTS` over other model-agnostic approaches for addressing distribution shifts, we perform experiments comparing its performance against distribution shift baselines,

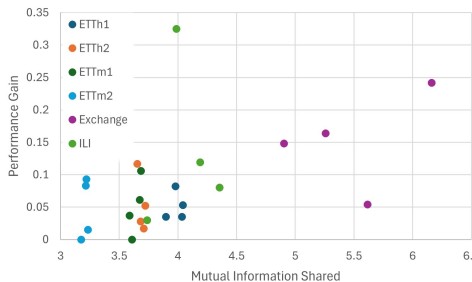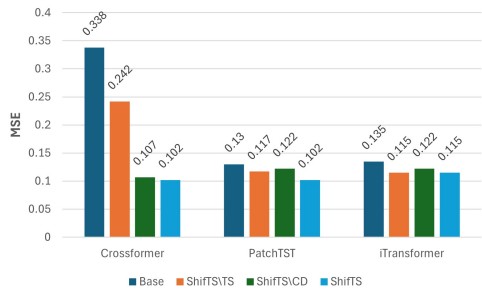

Figure 3: **Left (a):** The performance gains of `ShifTS` versus the mutual information shared between $\mathbf{X}^H$ and $\mathbf{Y}^H$. Greater mutual information in $\mathbf{X}^H$ compared to $\mathbf{Y}^H$ correlates with more significant performance gains achieved by `ShifTS`. **Right (b): Ablation Study.** Addressing either concept drift or temporal shift individually provides certain benefits in forecasting accuracy. `ShifTS` that tackles both achieves the lowest forecasting error.

including methods designed for concept drift, temporal shift, and combined approaches. We exclude evaluations on minutely ETT datasets, following (Liu et al., 2024a), as their data characteristics and forecasting performance closely resemble those of hourly ETT datasets. The experiments utilize Crossformer as the forecasting model, and the averaged results are presented in Table 2.

The results highlight the advantages of `ShifTS` over existing distribution shift methods, achieving the highest average forecasting accuracy in 6 out of 8 evaluations, with the remaining 2 evaluations ranking second. Notably, as discussed in Section 3.2, we choose to use RevIN as it is one of the most popular yet simple and effective temporal shift methods. However, `ShifTS` is flexible and can integrate more advanced temporal shift methods to further enhance performance. While exploring these advanced temporal shift methods is beyond the scope of this work, we illustrate the potential benefits of such integration. For example, on the Exchange dataset, where SAN outperforms `ShifTS`, incorporating SAN in place of RevIN within `ShifTS` leads to even greater accuracy improvements. Detailed MSE values for these evaluations are provided in Table 3.

Table 3: MSE comparison between `ShifTS`, SAN, and `ShifTS`+SAN on Exchange dataset. `ShifTS`+SAN achieves the best performance on all evaluations.

| Horizon | `ShifTS` | SAN | `ShifTS` w. SAN |
|---------|----------|-------|-----------------|
| 96      | 0.102    | 0.091 | **0.089**       |
| 192     | 0.207    | 0.195 | **0.187**       |
| 336     | 0.407    | 0.373 | **0.372**       |
| 720     | 1.165    | 1.001 | **0.981**       |
| Avg.    | 0.470    | 0.415 | **0.407**       |

Furthermore, the results underscore the importance of addressing concept drift using SAM when temporal shifts are effectively addressed.

## 4.4 ABLATION STUDY

To demonstrate the effectiveness of each module in `ShifTS`, we conducted an ablation study using two modified versions: `ShifTS`\TS and `ShifTS`\CD. `ShifTS`\TS excludes the temporal shift adjustment via RevIN, while `ShifTS`\CD excludes the concept drift handling via SAM. Additionally, conventional forecasting models that do not address either concept drift or temporal shift are denoted as 'Base'. We performed experiments on the Exchange datasets using the previous three baseline forecasting models, with a fixed forecasting horizon of 96. The results are visualized in Figure 3(b). The visualization reveals the following observations:

First, addressing temporal shift and concept drift together, as implemented in `ShifTS`, yields lower forecasting errors than addressing only one type of distribution shift (`ShifTS`\TS and `ShifTS`\CD) or not considering any distribution shift adjustments (Base). This suggests that temporal shift and concept drift are interrelated and co-exist in time series data, and addressing both provides significant benefits. Second, for forecasting models that inherently address temporal shift, such as PatchTST and

iTransformer that incorporate norm/denorm, the performance gains from mitigating concept drift are more significant than those from additionally mitigating temporal shift using RevIN. In contrast, for models without any temporal shift mitigation, such as Crossformer, tackling temporal shift leads to a greater performance improvement than concept drift. These observations suggest that mitigating temporal shift is a necessity in mitigating concept drift, which matches the intuition in Section 3.2.

## 5 CONCLUSION AND LIMITATION DISCUSSION

In this paper, we identify the challenges posed by both concept drift and temporal shift in time-series forecasting. While the issue of mitigating temporal shifts has garnered significant attention within the time-series forecasting community, concept drift has remained largely overlooked. To bridge this gap, we propose SAM, a method designed to effectively address concept drift in time-series forecasting by modeling conditional distributions through surrogate exogenous features. Building on SAM, we introduce ShifTS, a model-agnostic framework that handles concept drift in practice by first mitigating temporal shift as a preliminary step. Our comprehensive evaluations highlight the effectiveness of ShifTS, while the benefits of SAM are further demonstrated through an ablation study. We discuss the limitations of our approach in Appendix E.

## 6 ACKNOWLEDGMENT

This work was partly supported by the NSF (Expeditions CCF-1918770, CAREER IIS-2028586, Medium IIS-1955883, Medium IIS-2403240, Medium IIS-2106961), NIH (1R01HL184139), CDC MInD program, Meta, and Dolby faculty gifts.

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

## A   RELATED WORKS

**Time-Series Forecasting.** Recent works in deep learning have achieved notable achievements in time-series forecasting, such as RNNs, LSTNet, N-BEATS (Sherstinsky, 2020; Lai et al., 2018; Oreshkin et al., 2020). State-of-the-art models build upon the successes of self-attention mechanisms (Vaswani et al., 2017) with transformer-based architectures and significantly improve forecasting accuracy, such as Informer, Autoformer, Fedformer, PatchTST, iTransformer, FRNet (Zhou et al., 2021; Wu et al., 2021; Zhou et al., 2022; Nie et al., 2023; Liu et al., 2024d; Zhang et al., 2024; Zhao et al., 2025c). However, these advanced models primarily rely on empirical risk minimization (ERM) with IID assumptions, i.e., train and test dataset follows the same data distribution, which exhibits limitations when potential distribution shifts in time series (Zhao et al., 2025b).

**Distribution Shift in Time-Series Forecasting.** In recent decades, learning under non-stationary distributions, where the target distribution over instances changes with time, has attracted attention within learning theory (Kuh et al., 1990; Bartlett, 1992). In the context of time series, the distribution shift can be categorized into concept drift and temporal shifts.

General concept drift methods (via invariant learning) (Arjovsky et al., 2019; Ahuja et al., 2021; Krueger et al., 2021; Pezeshki et al., 2021; Sagawa et al., 2019) assume instances sampled from various environments and propose to identify and utilize invariant predictors across these environments. However, when applied to time-series forecasting, these methods encounter limitations. Additional methods specifically tailored for time series data also encounter certain constraints: DIVERSITY (Lu et al., 2023) is designed for time series classification and detection only. OneNet (Wen et al., 2024) is tailored solely for online forecasting scenarios using online ensembling. PeTS (Zhao et al., 2025a) focuses on distribution shifts induced by the specific phenomenon of performativity.

Other works explicitly address temporal distribution shift in time-series forecasting (Kim et al., 2021; Liu et al., 2022; Fan et al., 2023; Liu et al., 2023). These methods typically design normalization schemes that align the statistical properties of the lookback and forecast horizon, ensuring that both segments follow comparable normalized distributions. Such alignment mitigates temporal shift arising from discrepancies between historical observations and future targets. More recent time-series foundation models Aksu et al. (2024); Woo et al. (2024); Das et al. (2024); Liu et al. (2025b); Ansari et al. (2025) improve generalization by scaling model capacity and training on large and diverse corpora. Through large-scale pretraining and parameter scaling, these approaches may implicitly enhance robustness to distribution shift. However, most existing foundation models focus primarily on univariate settings and do not explicitly leverage rich exogenous information. In parallel, several recent studies (Liu et al., 2024b;c; Liu et al.; 2025a; 2026) incorporate auxiliary textual context to assist time-series understanding and forecasting. These approaches extend beyond purely numerical modeling and explore multimodal conditioning, which is orthogonal to the scope of this work.

## B   TEMPORAL SHIFT AND CONCEPT DRIFT

To highlight the differences between concept drift and temporal shift, we provide visualizations of both phenomena. Figure 4 illustrates temporal shift, while Figure 5 demonstrates concept drift[2].

---

[2]Figures adapted from: `https://github.com/ts-kim/RevIN`

Temporal shift refers to changes in the statistical properties of a univariate time series data, such as mean, variance, and autocorrelation structures, over time. For instance, the mean and variance of the given time series shift between the lookback window and horizon window, as depicted in Figure 4. This issue is inherent in time series forecasting and can occur on any given time series data, regardless of whether the data pertains to the target series or exogenous features.

In contrast, concept drift describes to changes in the correlations between exogenous features and the target series over time. Figure 5 illustrates this phenomenon, where increases in exogenous features at earlier time steps lead to increases in the target series, while increases at later time steps result in decreases. Unlike temporal shift, concept drift involves multiple correlated time series and is not an inherent issue in univariate time series analysis.

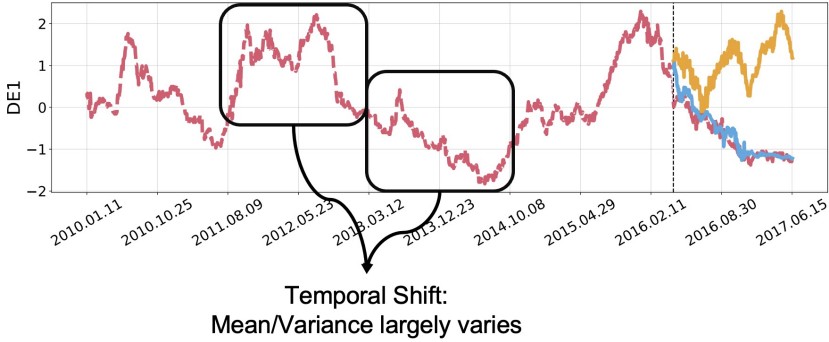

Figure 4: Demonstration of temporal shift phenomenon within time series data, showcasing the variations in statistical properties, including mean and variance, over time as the emergence of temporal shift (**Red:** ground truth; **Yellow:** N-BEATS prediction; **Blue:** N-BEATS+RevIN prediction).

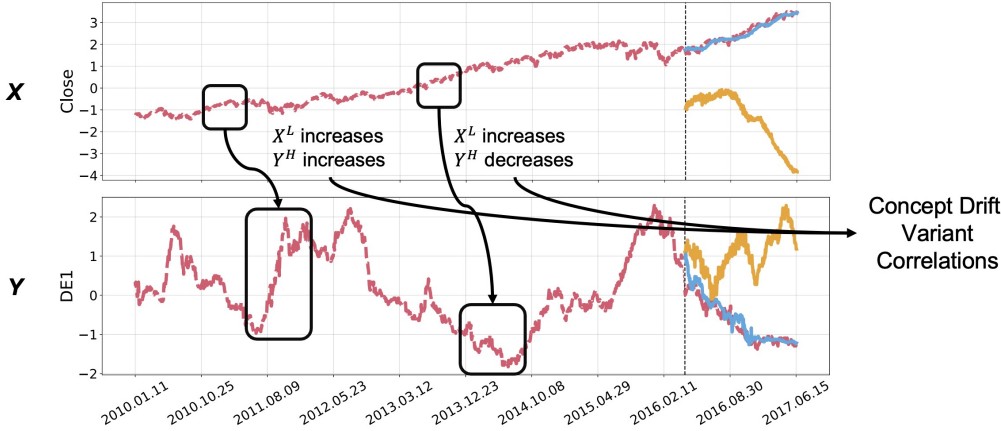

Figure 5: Demonstration of concept drift phenomenon within time series data, showcasing the variations in correlation structures between target series $\mathbf{Y}$ and exogenous feature $\mathbf{X}$ over time as the emergence of concept drift (**Red:** ground truth; **Yellow:** N-BEATS prediction; **Blue:** N-BEATS+RevIN prediction).

## C  ADDITIONAL EXPERIMENT DETAILS

### C.1  DATASETS

We conduct experiments on six real-world datasets, which are commonly used as benchmark datasets:

- **ILI.** The ILI dataset collects data on influenza-like illness patients weekly, with eight variables.

- **Exchange.** The Exchange dataset records the daily exchange rate of eight currencies.

- **ETT.** The ETT dataset contains four sub-datasets: **ETTh1**, **ETTh2**, **ETTm1**, **ETTm2**. The datasets record electricity transformer temperatures from two separate counties in China (distinguished by '1' and '2'), with two granularities: minutely and hourly (distinguished by 'm' and 'h'). All sub-datasets have seven variables/features.

We follow (Wu et al., 2022; Nie et al., 2023; Liu et al., 2024d) to preprocess data, which guides splitting datasets into train/validation/test sets and selecting the target variables. All datasets are preprocessed using the zero-mean normalization method.

Additional popular time-series datasets, such as Traffic (which records road occupancy rates from various sensors on San Francisco freeways), Electricity (which tracks hourly electricity consumption for 321 customers), and Weather (which collects 21 meteorological indicators in Germany, such as humidity and air temperature), are omitted from our evaluations. These datasets exhibit strong periodic signals and display near-stationary properties, making distribution shift issues less prevalent. A visualization comparison between the ETTh1 and Traffic datasets, shown in Figure 6, further supports this observation.

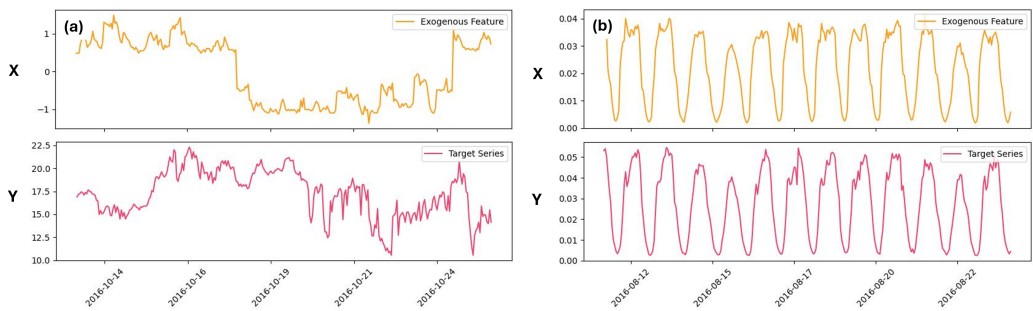

Figure 6: Distribution shift issues across datasets: **Left (a): ETT.** Both temporal shift and concept drift are present. The target series shows varying statistics over time (e.g., lower variance in earlier periods and higher variance later), causing temporal shift. The correlation between **X** and **Y** is unclear and unstable, causing concept drift. **Right (b): Traffic.** Both temporal shift and concept drift are moderate. The target series exhibits near-periodicity, making the temporal shift moderate. Moreover, the correlation between **X** and **Y** remains stable (e.g., both increase or decrease simultaneously), making concept drift moderate.

## C.2 BASELINE IMPLEMENTATION

We follow the commonly adopted setup for defining the forecasting horizon window length, as outlined in prior works (Wu et al., 2022; Nie et al., 2023; Liu et al., 2024d). Specifically, for datasets such as ETT and Exchange, the forecasting horizon windows are chosen from the set [96, 192, 336, 720], with a fixed lookback window size of 96 and a consistent label window size of 48 for the decoder (if required). Similarly, for the weekly reported ILI dataset, we employ forecasting horizon windows from [24, 36, 48, 60], with a fixed lookback window size of 36 and a constant label window size of 18 for the decoder (if required).

In the context of concept drift baselines, several baselines like GroupDRO, IRM, and VREx require environment labels, which are typically absent in time series datasets. To address this, we partition the training set into $k$ equal-length time segments to serve as predefined environment labels.

For baseline time-series forecasting models, we follow implementations and suggested hyperparameters (with additional tuning) sourced from the Time Series Library[3]. For concept drift baselines, we utilize implementations and hyperparameter tuning strategies recommended by DomainBed[4]. For temporal shift baselines, we adopt implementations and hyperparameter configurations outlined

---

[3]https://github.com/thuml/Time-Series-Library
[4]https://github.com/facebookresearch/DomainBed

in their respective papers. Additionally, we add an additional MLP layer to the end PatchTST to effectively utilize exogenous features, following (Liu et al., 2024a).

In the ablation study, for the implementation of PatchTST and iTransformer, we follow the original approach by applying norm and denorm operations to the 'Base' model. To clarify our notation, `ShifTS\TS` refers to the model with standard norm/denorm operations and `SAM`, while `ShifTS\CD` denotes the version where the regular norm/denorm is replaced with RevIN.

### C.3 Mutual Information Visualization

For a given time series dataset, we compute the mutual information $I(\mathbf{X}^H; \mathbf{Y}^H)$ for each training time step and each exogenous feature dimension individually, following:

$$I(\mathbf{X}^H; \mathbf{Y}^H) = \sum_{x \in \mathbf{X}^H} \sum_{y \in \mathbf{Y}^H} P(x, y) \log \frac{P(x, y)}{P(x)P(y)} \tag{5}$$

We then average the mutual information across all time steps for each exogenous feature dimension and identify the maximum averaged mutual information over all feature dimensions. This process allows us to assess the information content of each feature dimension in relation to the target series.

We visualize the maximum averaged mutual information plotted against the corresponding performance gain in Figure 3(a). This visualization provides insights into how the information content of different feature dimensions relates to the performance improvement achieved in the forecasting model.

## D Additional Results

### D.1 Evaluations on Agnostic Performance Gains

To further demonstrate the benefit of `ShifTS` in improving the forecasting accuracy over agnostic forecasting models, we additionally evaluate the performance differences without and with `ShifTS` on Informer, Pyraformer, and TimeMixer. The detailed results are presented in Table 4. The additional evaluations again show consistent performance improvements in these models. Moreover, compared to the results in Table 1, the performance gains on these older models are even more significant. This observation highlights the need to mitigate both concept drift and temporal shift in time-series forecasting, as such problem are rarely considered in these models, but in the later models (e.g., PatchTST and iTransformer are compounded with normalizaiton/denormalizaiton processes).

## E Limitation Discussion

This work introduces `SAM` to address concept drift and proposes an integrated framework, `ShifTS`, which combines `SAM` with temporal shift mitigation techniques to enhance the accuracy of time-series forecasting. Extensive empirical evaluations support the effectiveness of these methods. However, the limitations of this study lie in two aspects: First, the distribution shift methods in time-series forecasting, including `ShifTS`, lack a theoretical guarantee. For example, no analysis quantifies how much the error bound can be tightened by addressing concept drift or temporal shift compared to vanilla time-series forecasting methods. Second, while this paper defines concept drift and temporal shift issues within the context of time-series forecasting, `SAM` and `ShifTS` are not the only possible solutions. Exploring alternative approaches remains an avenue for future research beyond the scope of this work. These two limitations highlight opportunities for future investigation.

Table 4: Performance comparison on forecasting errors without (ERM) and with `ShifTS` on Informer, Pyraformer, and TimeMixer. Employing `ShifTS` again shows near-consistent performance gains agnostic to forecasting models. The top-performing method is in bold. 'IMP.' denotes the average improvements over all horizons of `ShifTS` vs ERM.

| Model | | Informer (AAAI'21) | | | | Pyraformer (ICLR'21) | | | | TimeMixer (ICLR'24) | | | |
|---|---|---|---|---|---|---|---|---|---|---|---|---|---|
| Method | | ERM | | ShifTS | | ERM | | ShifTS | | ERM | | ShifTS | |
| Dataset | | MSE | MAE | MSE | MAE | MSE | MAE | MSE | MAE | MSE | MAE | MSE | MAE |
| ILI | 24 | 5.032 | 1.935 | **1.030** | **0.812** | 4.692 | 1.898 | **0.979** | **0.749** | 0.853 | 0.733 | **0.789** | **0.702** |
| | 36 | 4.475 | 1.876 | **1.046** | **0.850** | 4.814 | 1.950 | **0.866** | **0.740** | 0.721 | 0.676 | **0.697** | **0.665** |
| | 48 | 4.506 | 1.879 | **0.918** | **0.818** | 4.109 | 1.801 | **0.789** | **0.732** | **0.737** | **0.692** | 0.741 | 0.711 |
| | 60 | 4.313 | 1.850 | **0.957** | **0.839** | 4.483 | 1.850 | **0.723** | **0.698** | 0.788 | 0.723 | **0.670** | **0.659** |
| | IMP. | | | 78.4% | 56.0% | | | 81.5% | 61.1% | | | 6.3% | 3.0% |
| Exchange | 96 | 0.839 | 0.746 | **0.137** | **0.277** | 0.410 | 0.525 | **0.145** | **0.275** | 0.127 | 0.268 | **0.098** | **0.234** |
| | 192 | 0.862 | 0.773 | **0.210** | **0.346** | 0.529 | 0.610 | **0.300** | **0.404** | 0.229 | 0.355 | **0.214** | **0.352** |
| | 336 | 1.597 | 1.063 | **0.378** | **0.485** | 0.851 | 0.778 | **0.440** | **0.506** | 0.553 | 0.560 | **0.440** | **0.491** |
| | 720 | 4.358 | 1.935 | **0.760** | **0.655** | 1.558 | 1.067 | 1.509 | **0.963** | 1.173 | 0.834 | **0.962** | **0.747** |
| | IMP. | | | 79.5% | 59.7% | | | 39.8% | 31.5% | | | 16.9% | 9.1% |
| ETTh1 | 96 | 0.891 | 0.863 | **0.095** | **0.231** | 0.653 | 0.748 | **0.065** | **0.197** | **0.059** | **0.184** | **0.059** | 0.187 |
| | 192 | 1.027 | 0.958 | **0.096** | **0.237** | 0.853 | 0.828 | **0.075** | **0.210** | 0.099 | 0.247 | **0.077** | **0.211** |
| | 336 | 1.055 | 0.961 | **0.092** | **0.237** | 0.705 | 0.797 | **0.092** | **0.238** | 0.121 | 0.279 | **0.098** | **0.246** |
| | 720 | 1.077 | 0.969 | **0.100** | **0.252** | 0.562 | 0.695 | **0.126** | **0.279** | 0.139 | 0.299 | **0.099** | **0.252** |
| | IMP. | | | 90.7% | 74.5% | | | 86.4% | 69.6% | | | 23.3% | 10.1% |
| ETTh2 | 96 | 3.195 | 1.651 | **0.232** | **0.381** | 1.598 | 1.127 | **0.156** | **0.307** | 0.152 | 0.303 | **0.146** | **0.299** |
| | 192 | 3.569 | 1.778 | **0.334** | **0.464** | 3.314 | 1.599 | **0.217** | **0.367** | 0.195 | 0.349 | **0.185** | **0.343** |
| | 336 | 2.556 | 1.468 | **0.400** | **0.512** | 2.571 | 1.489 | **0.245** | **0.398** | 0.238 | 0.392 | **0.230** | **0.381** |
| | 720 | 2.723 | 1.532 | **0.489** | **0.579** | 2.294 | 1.409 | **0.261** | **0.410** | 0.273 | 0.421 | **0.249** | **0.397** |
| | IMP. | | | 82.0% | 69.5% | | | 90.6% | 73.5% | | | 5.3% | 2.9% |
| ETTm1 | 96 | 0.320 | 0.433 | **0.055** | **0.175** | 0.130 | 0.298 | **0.028** | **0.125** | 0.030 | 0.128 | **0.029** | **0.126** |
| | 192 | 0.459 | 0.582 | **0.079** | **0.211** | 0.240 | 0.4112 | **0.045** | **0.162** | **0.047** | 0.165 | **0.047** | **0.164** |
| | 336 | 0.457 | 0.556 | **0.104** | **0.243** | 0.359 | 0.512 | **0.062** | **0.192** | 0.063 | 0.191 | **0.060** | **0.189** |
| | 720 | 0.735 | 0.760 | **0.148** | **0.294** | 0.657 | 0.750 | **0.091** | **0.231** | 0.083 | 0.223 | **0.081** | **0.220** |
| | IMP. | | | 80.7% | 60.3% | | | 82.2% | 62.6% | | | 2.3% | 1.1% |
| ETTm2 | 96 | 0.191 | 0.345 | **0.154** | **0.298** | 0.275 | 0.422 | **0.075** | **0.200** | 0.079 | 0.205 | **0.075** | **0.201** |
| | 192 | 0.458 | 0.556 | **0.243** | **0.378** | 0.484 | 0.552 | **0.107** | **0.248** | 0.121 | 0.259 | **0.111** | **0.250** |
| | 336 | 0.606 | 0.624 | **0.515** | **0.539** | 1.138 | 0.909 | **0.146** | **0.293** | 0.150 | 0.295 | **0.148** | **0.294** |
| | 720 | 1.175 | 0.879 | **0.564** | **0.592** | 2.920 | 1.537 | **0.196** | **0.347** | 0.246 | 0.387 | **0.198** | **0.346** |
| | IMP. | | | 33.4% | 23.0% | | | 82.8% | 63.2% | | | 8.5% | 4.1% |

