# OpenReview forum: "Tackling Time-Series Forecasting Generalization via Mitigating Concept Drift"
_ICLR.cc/2026/Conference — ICLR 2026 Poster_

### Official Review · Reviewer_YbKf · 2025-10-25

**Soundness:** 3
**Presentation:** 3
**Contribution:** 3
**Rating:** 6
**Confidence:** 4

**Summary:**

This paper identifies two types of distribution shifts in time series forecasting: temporal shift (marginal distribution) and concept drift (conditional distribution). The authors propose a new method, SAM (soft attention masking), to mitigate concept drift by learning invariant patterns from both lookback and *horizon* exogenous features ($X^L$ and $X^H$). This mechanism learns a surrogate feature, $X^{SUR}$, which is predicted as an auxiliary task. The paper also presents ShifTS, a model agnostic framework that first uses normalization (like RevIN) to handle temporal shift, and then uses a backbone model to jointly predict the target $Y^H$ and the surrogate $\hat{X}^{SUR}$. Experiments on six datasets show that ShifTS consistently improves the performance of various backbone models, including current state of the art ones.

**Strengths:**

- The paper clearly distinguishes between temporal shift and concept drift, tackling a significant and practical problem in time series forecasting.
- The core idea of using *horizon* exogenous features ($X^H$) to define a stable surrogate target ($X^{SUR}$) is novel. This surrogate acts as an effective regularization target during training, forcing the model to learn future relevant patterns.
- The ShifTS framework is practical and model agnostic. It cleanly integrates a known temporal shift solution (normalization) with the proposed concept drift solution (SAM).
- The experiments are comprehensive. They show consistent performance improvements across six datasets and multiple strong backbone models, and also outperform other distribution shift baselines.

**Weaknesses:**

- The claim that SAM finds "invariant patterns" is not well supported by the mechanism. The method (Equation 1) is a learnable attention mask, not an explicit invariance optimization like in IRM. It seems to learn a *useful compression* of future features, but calling it "invariant" is a strong claim that needs better justification.
- The paper introduces significant complexity with the SAM module and the aggregation MLP. However, the performance gains on SOTA models like iTransformer are sometimes small (e.g., <5% MAE gain on ETTh2/ETTm2). The marginal benefit versus the added complexity is questionable in these cases.
- A critical and much simpler baseline is missing. The paper argues predicting raw $X^H$ is too hard, but $X^{SUR}$ is easier. This assertion must be tested by comparing ShifTS to a simpler multi task model that just predicts the raw $X^H$ as the auxiliary task.

**Questions:**

1.  Can you please clarify how the SAM mechanism (Equation 1) specifically enforces invariance? The high weight patterns are defined as invariant, but it is unclear how the optimization process encourages this property over just learning a stable predictive signal.
2.  To justify the complexity of SAM, could you add a baseline that replaces the $X^{SUR}$ target with the raw $X^H$? This would involve a multi task loss $\mathcal{L} = \mathcal{L}_{TS} + \lambda \cdot MSE(X^H, \hat{X}^H)$. This comparison is essential to prove that the SAM slicing and attention is superior to just predicting the raw future features.
3.  What is the impact of applying ShifTS to "near-stationary" datasets like Traffic or Weather, which were excluded? Does the method degrade performance in the absence of significant shifts, or does it robustly "do no harm"?
4.  The aggregation step ($Agg(\cdot)$ in Algorithm 1 and Figure 2) appears to be a key component. Could you provide more detail on its architecture and how the final $\hat{Y}^H$ is computed from the initial forecast and the $\hat{X}^{SUR}$ prediction?

---

> ### Author Response · Authors · 2025-11-18
> **Rebuttal to Reviewer YbKf**
>
> We appreciate the valuable comments from the reviewer, and we are willing to address the concerns.
>
> **Response to W1 & Q1: Invariant**
>
> Thank you for the comment. We agree that “invariant patterns” is a term often associated with domain adaptation settings, where explicit domain labels exist and methods such as IRM perform optimization to enforce domain-level invariance. In contrast, time-series forecasting rarely provides domain annotations, making explicit invariance optimization infeasible in our setting.
>
> Our intention is therefore different: SAM does not enforce invariance in the IRM sense, but instead aims to identify stable conditional relationships that persist across different temporal slices (lines 195–214). First, we construct sliding windows over both the look-back and horizon intervals, and compute the conditional distributions associated with each temporal slice. Then, at each time step, SAM produces a learnable weighting over these conditional distributions. By averaging these weights over time, we obtain a measure of how consistently each conditional distribution contributes to prediction across the entire time series. Conditional distributions that receive consistently high weights are those that remain useful across many time steps, reflecting temporal stability rather than domain-level invariance. Our use of the term “invariant” refers to this stability across temporal contexts, not IRM-style enforced invariance.
>
> **Response to W2: Performance Gain and Cost**
>
> As discussed in lines 408–416, we acknowledge that when ShifTS is applied to more advanced models such as iTransformer, the average improvement may appear smaller. However, we emphasize that the gains are still substantial on several datasets. For example, on ILI and Exchange, ShifTS delivers over 10% improvement, which is highly significant even for strong baselines. More importantly, the bottom line is that ShifTS does not degrade model performance, while still offering measurable improvements even on the advanced baselines.
>
> Regarding computational overhead, ShifTS adds only a small number of parameters. The training cost increase is reasonable; for example, on ETTh1 with iTransformer, the per-epoch time changes only from 3.6s to 6.3s with iTransformer on ETTh1-96-96 (15.1s to 18.5s with Crossformer). This indicates that the method delivers performance gains at very low additional complexity, making it practical even when the improvements are modest.
>
> **Response to W3 & Q2: Multi-task Baseline**
>
> Thank you for the suggestion. We conducted an additional experiment that incorporates the proposed multi-task baseline. We report the results at https://anonymous.4open.science/r/shifts_iclr-ED40/multi.pdf. The findings show that ShifTS consistently outperforms the multi-task baseline, particularly with longer horizons. This supports our intuition that predicting the entire future exogenous sequence is almost as difficult as forecasting the target itself. In contrast, SAM selectively focuses on the subsets of horizon information that contribute to stable conditional relationships, rather than attempting to reconstruct the full future feature trajectory.
>
> **Response to Q3: Near-Stationary Performance**
>
> Thanks for the suggestion. We have added experiments on the Weather datasets at https://anonymous.4open.science/r/shifts_iclr-ED40/weather.pdf. We observe that even for near-stationary time series, ShifTS will at least, and importantly, it does not harm performance, and may provide marginal improvements. This indicates that ShifTS remains robust even when temporal or conditional shifts are minimal.
>
> **Response to Q4: Aggregation**
>
> The Agg() is a simple dense layer, similar to a projection layer, that aggregates the $X^{SUR}$ across all exogenous feature dimensions to predict the target.

---

> > ### Comment · Reviewer_YbKf · 2025-11-19
> >
> > Thank you very much for the author's detailed reply and supplementary experiments, which confirm that my understanding of the article was correct. I have no further questions and will maintain my original rating. I wish the author all the best in the ICLR process.

---

### Official Review · Reviewer_ytQx · 2025-10-30

**Soundness:** 2
**Presentation:** 3
**Contribution:** 2
**Rating:** 6
**Confidence:** 3

**Summary:**

This paper categorizes the general concept drift in time series into types: concept drift and temporal shift (as defined more precisely in Definition 2.1 and Definition 2.2). To solve the concept drift problem, it proposes the soft attention mechanism (SAM) to find the invariant patterns in lookback and horizon windows. The core idea of this paper is illustrated well in Fig. 1. A method-agnostic framework called ShiftTS is proposed to deal with both temporal and concept drifts in a unified framework. Experiments demonstrate the good performance of the proposed method.

**Strengths:**

1.	The paper is well written and easy to follow. The idea proposed is simple.
2.	The invariant patterns are learned through the surrogate feature $X_{SUR}$ and this is the core contribution of this paper. The basic idea is to concatenate the lookback and horizon windows, and model the conditional distributions for local patterns using the soft attention matrix M.
3.	Experiments demonstrate the good performance of the proposed method.

**Weaknesses:**

1.	The categorization of temporal shift and concept drift is very similar to different sources of concept outlined in Section 2.1 in [R1], but in the context of time series with some differences (the temporal shift is for Y, and the concept drift is the same as Source II in [R1] ).
2.	The method mitigating temporal shift (or the marginal distribution shift of Y) looks quite standard in the literature.
3.	The proposal of mitigating concept drift may be incremental for some datasets and base algorithms.

**Questions:**

1.	The analysis in Section 4.2 looks reasonable to me. The improvement of the proposed method depends on the data and the base algorithm simultaneously. Fig.3(a) plots the performance gain vs. the mutual information between X and Y. In fact, this plot studies the effectiveness of ShiftTS w.r.t. the so-called concept drift defined in Def 2.2 in this paper. A good example is the Exchange dataset, which shows low mutual information but high performance improvement. Thus, it suggests that the performance is mainly due to the mitigation of the so-called temporal concept drift. This is also consistent of Fig. 3(b), which shows that the most significant performance improvement is between ShiftTS\TS and Base. Therefore, at least on the Exchange dataset, the proposal of learning form surrogate feature $X_{SUR}$ looks incremental.

---

> ### Author Response · Authors · 2025-11-18
> **Rebuttal to Reviewer ytQx**
>
> We appreciate the valuable comments from the reviewer, and we are willing to address the concerns.
>
> **Response to W1: Categorization**
>
> We agree that distribution shift has been extensively studied for decades, and therefore our definitions naturally align with prior formulations in general machine learning. From this perspective, the terminology itself is not intended as a novel theoretical contribution. However, we emphasize that very few time-series forecasting studies explicitly differentiate these two forms of shift. In our work, we not only make this distinction explicit in the time-series context but also provide real-world examples illustrating how temporal shift and concept drift are different in practice through visualization (e.g., Figures 4 & 5 on page 14). This conceptual understanding is important to appear in the problem definition section because it directly motivates the design of ShifTS and helps practitioners understand when and why the method is beneficial.
>
> **Response to W2: Standard Temporal Shift Method**
>
> We acknowledge that the temporal-shift mitigation techniques used in ShifTS are standard in the literature. However, we would like to emphasize that proposing a new method for mitigating temporal shift is not the main contribution of our work. Our primary focus is on addressing concept drift, and Section 3.2 is intended to clarify why mitigating temporal shift is a necessary prerequisite before one can effectively mitigate concept drift. This motivation is essential for understanding the design of the ShifTS framework. Thus, while the temporal-shift component itself is not novel, the insight into its necessity and its role within a broader concept-drift mitigation framework is both valuable and, we believe, novel.
>
> **Response to W3 & Q1: Incremental Gains**
>
> Thank you for the insightful comment. We agree that, in some cases, the performance improvements of ShifTS appear incremental. However, we believe this is common and understandable that a rare single forecasting method can consistently outperform all baselines across all datasets and model architectures by far, and moreover, we are aiming to mitigate concept drift issues via ShifTS for agnostic models, rather than beating the SOTA.
>
> Nevertheless, our results show that the bottom line of ShifTS does not degrade performance even in scenarios where the gain is modest. And more importantly, we explain and identify when ShifTS is expected to be effective by linking its performance to a measurable data property. This provides practitioners with a practical guideline to assess the potential benefits of ShifTS on their own datasets, rather than relying on universal dominance.

---

### Official Review · Reviewer_8foj · 2025-10-31

**Soundness:** 3
**Presentation:** 3
**Contribution:** 3
**Rating:** 6
**Confidence:** 4

**Summary:**

This paper addresses the problem of **distribution shifts in time-series forecasting**, focusing specifically on **concept drift**—a relatively under-explored issue compared to temporal shift. The authors propose:

1. **SAM**: A mechanism to identify invariant patterns in exogenous features across lookback and horizon windows, enabling more stable conditional distribution modeling.
2. **ShiftS**: A unified, model-agnostic framework that first mitigates temporal shift (via normalization) and then concept drift (via SAM), improving generalization across diverse forecasting models.

Extensive experiments on six real-world datasets show that ShiftS consistently boosts forecasting accuracy across multiple base models and outperforms existing distribution-shift baselines.

**Strengths:**

- **Originality**: SAM is a novel approach to handling concept drift without relying on environment labels or online retraining.
- **Quality**: The method is well-designed, with careful attention to both theoretical motivation and practical implementation.
- **Clarity**: The problem formulation and methodology are clearly explained, and the experiments are thorough and convincing.
- **Significance**: The paper fills a clear gap in the literature and provides a practical tool for improving time-series forecasting under distribution shifts.

**Weaknesses:**

- **Theoretical Guarantees**: The method lacks theoretical analysis (e.g., error bounds or convergence guarantees), which is noted in the limitations but could strengthen the contribution.
- **Dependence on Horizon Exogenous Data**: SAM relies on $\mathbf{X}^H$ , which may not always be available or accurately predictable in practice. The paper addresses this via surrogate forecasting, but the impact of prediction error on final performance is not deeply analyzed.
- **Limited Scope**: The method is evaluated only on univariate forecasting with exogenous features. Its applicability to multivariate or purely endogenous settings remains unclear.

**Questions:**

1. How does SAM perform when $\mathbf{X}^H$ is not available during training or is highly noisy? Have you tested scenarios with missing or imperfect exogenous data?
2. Could SAM be adapted for online settings where concept drift occurs continuously? The paper criticizes online methods but does not explore whether SAM can be extended in that direction.
3. The mutual information analysis is insightful—have you considered using $ I(\mathbf{X}^H; \mathbf{Y}^H) $ as a criterion for applying SAM in practice?
4. The framework currently uses RevIN for temporal shift mitigation. Have you experimented with more advanced methods (e.g., SAN) in the full ShiftS pipeline, and if so, how do they compare?

---

> ### Author Response · Authors · 2025-11-18
> **Rebuttal to Reviewer 8foj**
>
> We appreciate the valuable comments from the reviewer, and we are willing to address the concerns.
>
> **Response to W1: Theoretical Gaurantee**
>
> We appreciate the reviewer for raising this point. We acknowledge that our work does not include a comprehensive theoretical analysis (e.g., error bounds or convergence guarantees). However, we do offer conceptual guidance, for example, illustrating the necessity of handling temporal shift before concept drift, or the trade-off between the difficulty of predicting farther-ahead exogenous features and the benefit they bring to forecasting the target series, and we empirically demonstrate that the benefits outweigh the drawbacks in practice.
>
> More importantly, we identify when ShifTS is likely to be beneficial through a simple mutual-information-based measurement. While this does not give a formal theoretical guarantee, it provides a practical and interpretable criterion for real-world usage, which we believe is valuable for practitioners. We believe theoretical analysis on ShifTS is a promising future research study, but currently out of our scope.
>
> **Response to W2 & Q1: Availability and Quality of Exogenous Data**
>
> Thank you for raising this important point. We clarify that the unobserved target-aligned exogenous features $X^H$ are not missing during training in typical forecasting pipelines. For example, in exchange rate prediction, all historical exogenous features at all historical time steps are fully observed. By definition, $X^H$ corresponds to the exogenous inputs at the same time steps as the training labels $Y^H$. Since both belong to the historical portion of the time series, they are naturally available and do not require forecasting during training.
>
> For data quality, we evaluate ShifTS on datasets such as ETT, which are well known for their irregular and noisy time-series patterns. The empirical results show that ShifTS consistently improves forecasting accuracy even under such imperfect conditions, suggesting robustness to realistic levels of noise.
>
> **Response to W3: Limited Scope**
>
> Thank you for raising this point. However, we respectfully disagree with the characterization that our scope is limited. First, univariate forecasting with exogenous features (often referred to as multi-single forecasting) is itself a widely used and practically important setting. Many real-world applications, such as epidemic forecasting or industrial demand forecasting, rely on predicting a single target series supported by multiple external signals, which aligns directly with our setup. Second, ShifTS can be potentially extended to multivariate forecasting. One can view each output dimension as a target series and treat the remaining dimensions as exogenous inputs, applying ShifTS separately to each dimension. This provides a straightforward path toward a multivariate variant of our framework. While such an extension is promising, it is currently beyond our primary scope and thus remains for future work.
>
> **Response to Q2: Online Setting**
>
> Thank you for the question. Our work focuses on the standard offline supervised forecasting setup. However, ShifTS can be extended to online or real-time setup. As new data arrive, practitioners could incrementally update the forecasting model and dynamically adjust the learned weights in SAM. We agree this is an interesting direction and will include a discussion in the final version, though it currently lies outside the evaluation scope of this work.
>
> **Response to Q3: Mutual Information**
>
> Thank you for the thoughtful question. Our results indeed show a strong and statistically significant correlation between mutual information (MI) and the performance gains of ShifTS. A higher MI generally indicates larger improvements. However, using MI as a hard decision criterion for when to apply SAM would require additional theoretical analysis to fully support the relationship between MI and conditional stability, which we view as a promising direction for future work but beyond the current scope. Practically, ShifTS holds the “do-no-harm’’ property: it does not degrade performance even when MI is low, while offering gains when MI is high. Since the added computational cost is relatively marginal (e.g., 15.1s to 18.5s per-epoch with Crossformer on ETTh1-96-96), practitioners can safely apply ShifTS without requiring MI-based gating, though MI can still serve as a useful heuristic indicator of potential benefit.
>
> **Response to Q4: ShifTS with SAN**
>
> Thanks, we are able to provide additional experiments that replace RevIN with SAN on Exhcange and ETTh2 with Crosformer and iTransformer at https://anonymous.4open.science/r/shifts_iclr-ED40/wsan.pdf, which show consistent results as Table 3, where ShifTS improves further with more advanced temporal shift methods.

---

### Meta-Review · Area_Chair_Rqi1 · 2026-01-10

**Summary:**

This paper studies generalization issues in time-series forecasting under distribution shifts and makes a clear distinction between temporal shift and concept drift. The authors argue that while temporal shift has been widely studied, concept drift in time-series forecasting remains comparatively underexplored. To address this, this paper introduces soft attention masking (SAM) that identifies stable patterns from both lookback and horizon exogenous features, and further introduces ShifTS, a model-agnostic framework that mitigates temporal shift first and then concept drift in a unified pipeline.

The reviewers agree that the paper is well written, technically sound, and empirically solid. All reviewers rate the paper marginally above the acceptance threshold. Overall, this work presents a coherent framework that makes a meaningful contribution to time-series forecasting under distribution shift.

**Reviewer Concerns:**

Several concerns were raised regarding theoretical guarantees, limited scope, the significance given existing literature, and empirical analysis. While two reviewers did not respond, the concerns from Reviewer YbKf have been largely addressed. Regarding theoretical analysis, I agree that analyzing the generalization performance for time-series forecasting (especially under distribution shifts) is fundamentally challenging, and is out of the scope of this work.

**Reviewer Scores:**

Reviewer 8foj: Rated the paper marginally above the acceptance threshold, but did not participate in the discussion.

Reviewer ytQx: Also rated the paper marginally above the threshold, expressing some reservations about novelty and incremental gains. The rebuttal directly addressed these points, and while skepticism remains, no major unresolved technical issues persist.

Reviewer YbKf: Provided a positive assessment on soundness, clarity, and practical relevance, with a score above the threshold. Clarifications in the rebuttal adequately addressed concerns about invariance claims and complexity.

---

### Decision · Program_Chairs · 2026-01-26

Accept (Poster)